# Tauroursodeoxycholic Acid Enhances Osteogenic Differentiation through EGFR/p-Akt/CREB1 Pathway in Mesenchymal Stem Cells

**DOI:** 10.3390/cells12111463

**Published:** 2023-05-24

**Authors:** Hyojin Kang, Sunsik Yang, Jun Lee

**Affiliations:** 1Department of Oral and Maxillofacial Surgery, College of Dentistry, Wonkwang University, 77 Dunsan-ro, Seo-gu, Daejeon 35233, Republic of Korea; khj5797@naver.com; 2Bonecell Biotech Inc., 77 Dunsan-dong, Seo-gu, Daejeon 35233, Republic of Korea; nessie6521@bonecell.co.kr

**Keywords:** tauroursodeoxycholic acid, osteogenic differentiation, epidermal growth factor receptor, human mesenchymal stem cell

## Abstract

Background: Mesenchymal stem cells (MSCs) are pluripotent stromal cells that are among the most appealing candidates for regenerative medicine and may aid in the repair and regeneration of skeletal disorders through multiple mechanisms, including angiogenesis, differentiation, and response to inflammatory conditions. Tauroursodeoxycholic acid (TUDCA) has recently been used in various cell types as one of these drugs. The mechanism of osteogenic differentiation by TUDCA in hMSCs remains unknown. Methods: Cell proliferation was performed by the WST-1 method, and alkaline phosphatase activity and alizarin red-sulfate staining were used to confirm the osteogenic differentiation indicator. Expression of genes related to bone differentiation and specific genes related to signaling pathways was confirmed by quantitative real-time polymerase chain reaction. Results: We found that cell proliferation was higher as the concentration increased, and showed that the induction of osteogenic differentiation was significantly enhanced. We also show that osteogenic differentiation genes were upregulated, with the expression of the epidermal growth factor receptor (EGFR) and cAMP responsive element binding protein 1 (CREB1) being specifically high. To confirm the participation of the EGFR signaling pathway, the osteogenic differentiation index and expression of osteogenic differentiation genes were determined after using an EGFR inhibitor. As a result, EGFR expression was remarkably low, and that of CREB1, cyclin D1, and cyclin E1 was also significantly low. Conclusions: Therefore, we suggest that TUDCA-induced osteogenic differentiation of human MSCs is enhanced through the EGFR/p-Akt/CREB1 pathway.

## 1. Introduction

Stem cells can self-renew and differentiate into specific cell types [1]. Mesenchymal stem cells (MSCs) are pluripotent stromal cells and are the most used stem cells in current preclinical and clinical studies of skeletal diseases through direct injection or scaffold (collagen, hyaluronic acid (HA)-based natural gels and hydrogels, gelatin, and alginate) transplantation [2,3,4,5]. MSCs can differentiate into various cell types, including osteoblasts, adipocytes, chondrocytes, and myocytes. Dexamethasone, insulin, and transforming growth factor beta 1 in a culture medium are used to induce differentiation into the desired cell type [6,7,8]. According to several in vitro and in vivo studies in regenerative medicine, MSCs are the most commonly used stem cells and are highly involved in the bone-healing process, as they have a very high potential to increase osteoinduction and osteogenesis. In other words, MSCs play an important role in bone repair and regeneration through several mechanisms, including angiogenesis, inflammatory responses, and promotion of cell migration and differentiation [9,10].

Recently described as a stem cell inducer, tauroursodeoxycholic acid (TUDCA) is a bile acid produced in small amounts in the human body that has been approved by the U.S. Food and Drug Administration (FDA) for safe use as a drug for human consumption [11,12]. TUDCA plays an effective role in liver disease [13], is involved in angiogenesis [12,14], has therapeutic effects on degenerative nervous system diseases, including Parkinson’s disease [15] and Alzheimer’s disease [16], and is effective in bone regeneration [17,18]. In addition, bone regeneration has been reported to occur by suppressing ER stress signals [19]. MSCs isolated from rat bone marrow have been explored for bone regeneration by TUDCA [17]. To date, the signaling mechanism of TUDCA-induced osteogenic differentiation remains unclear. In addition, there are few reports on osteogenic differentiation by TUDCA in human mesenchymal stem cells. We studied to search for specific genes and confirm their expression in order to find new signaling mechanisms.

Therefore, in this study, the osteogenic differentiation effect of TUDCA was confirmed in hMSC, and the signaling mechanism of osteogenic differentiation by TUDCA was identified at the molecular level through target gene expression.

## 2. Materials and Methods

### 2.1. Cell Culture and Treatment

hMSCs isolated from adult bone marrow (ATCC, Manassas, VA, USA), TUDCA (Sigma-Aldrich, Saint Louis, MO, USA), and an epidermal growth factor receptor (EGFR) inhibitor (AG1478; Sigma-Aldrich) were purchased and used in this study. The cells were cultured in an α-minimum essential medium (Gibco, Gaithersburg, MD, USA) containing 10% fetal bovine serum (Gibco) and 1% antibiotics (10,000 U/mL penicillin G and 25 μg/mL amphotericin B; Gibco) in a humidified CO_2_ incubator at 37 °C. Cells at passages 4–5 were used in this study.

### 2.2. Cell Proliferation Assay

Cells were seeded at a density of 5 × 10^3^ cells/well in a 96-well plate and treated with TUDCA at different concentrations. After culturing for days 1, 3, 5, and 7, the cells were washed twice with phosphate-buffered saline. EZ-cytox™ (Daeil Lab Service, Seoul, Republic of Korea), which has a method of analyzing cell proliferation by measuring the water-soluble tetrazolium salts (WST-1) reacted by dehydrogenase in mitochondria, is added to 10% of the culture medium and treated with cells and reacted for 1 h in a 37 °C incubator. After the reaction, the culture medium was analyzed by a microplate reader (SpectraMax M3; Molecular Devices, Sunnyvale, CA, USA), which was measured the absorbance at 450 nm.

### 2.3. Alkaline Phosphatase (ALP) Activity Assay

To measure ALP activity, cells were seeded at a density of 5 × 10^3^ cells/well in a 96-well plate and treated with TUDCA at different concentrations. After 7 d, the 96-well plates were washed twice with phosphate-buffered saline. The cells were resuspended in 0.1% Triton-X 100 (Sigma-Aldrich) at 4 °C for 1 h. The absorbance of the cell lysate was measured at 405 nm in a reaction with *p*-nitrophenyl phosphate (Sigma-Aldrich), and ALP activity was calculated using a standard formula. Protein quantification of cell lysates was performed using a TCA assay kit (Sigma-Aldrich). The amount of *p*-nitrophenyl phosphate was normalized to the total amount of protein and recorded as nM/μg protein.

### 2.4. Calcium Accumulation Measurement

Calcium accumulation was analyzed using alizarin red-sulfate (AR-S; Sigma-Aldrich) staining. Cells were seeded in 96-well plates at a density of 5 × 10^3^ cells/well and continuously cultured in TUDCA for 20 d. Cells were then fixed with 70% (*v*/*v*) ethanol for 1 h at 4 °C and stained with 40 mM AR-S solution. Cells were observed under a microscope (DM IL LED Fluo; Leica Microsystems, Wetzlar, Germany) at 2.5× magnification and solubilized using 10% (*w*/*v*) cetylpyridinium chloride in 10 mM sodium phosphate (pH 7.0). A microplate reader (SpectraMax M3) was used to measure the absorbance at 562 nm.

### 2.5. Quantitative Real-Time Polymerase Chain Reaction (qRT-PCR) Analysis and Western Blotting

To evaluate the effect of TUDCA on osteogenesis, the mRNA expression of several osteoblast marker genes and those involved in the relevant signaling pathway were determined. Cells were seeded at a density of 5 × 10^5^ cells/well in a 6-well plate and grown in a maintenance medium until confluence. The cells were then treated with 250 and 500 μM TUDCA in the presence or absence of EGFR inhibitors. Total RNA was extracted using a TRIzol reagent (Ambion, Carlsbad, CA, USA), and template DNA was prepared using a cDNA synthesis kit (LunaScript^®^ RT SuperMix Kit, NEB, Ipswich, MA, USA). Real-time gene expression amplification was performed using SYBR Green dye (Luna^®^ Universal qPCR Master Mix; NEB, USA). PCR assays were performed using a StepOnePlus Real-Time PCR System (Applied Biosystems, Branchburg, NJ, USA). The sequences of primers are shown in Appendix A.

For Western blotting, cells were seeded at a density of 5 × 10^5^ cells/well in 6-well plates and grown in a maintenance medium until confluence. The cells were then treated with 250 and 500 μM TUDCA in the presence or absence of EGFR inhibitors. Cell cultures were harvested at the times indicated and proteins extracted at 4 °C in a PRO-PREP solution (Intron Biotechnology, Seoul, Republic of Korea). Extracts were cleared by centrifugation, and protein concentrations were determined using a protein assay kit (Bio-Rad, Hercules, CA, USA). Proteins were separated by 12% sodium dodecyl sulfate–polyacrylamide gel electrophoresis (10 μg of each sample) and transferred onto nitrocellulose membranes. The blots were blocked with a buffer containing 0.05% Tween-20 and 1% bovine serum albumin and sequentially reacted with primary and secondary antibodies. Primary antibodies against p-Akt, Akt (Cell Signaling Technology, Beverly, MA, USA), and glyceraldehyde 3-phosphate dehydrogenase (GAPDH; Santa Cruz Biotechnology, Santa Cruz, CA, USA) were diluted to 1:1000. Horseradish peroxidase-conjugated secondary antibodies were diluted to 1:10,000. Finally, the membranes were developed using an enhanced chemiluminescence solution and exposed to an X-ray film.

### 2.6. Statistical Analysis

At least three independent sets of experiments were performed in triplicates for each condition. Statistical analysis between groups was confirmed by one-way analysis of variance (ANOVA) and Dunnett’s post hoc test using GraphPad Prism 6 statistical analysis software (GraphPad, San Diego, CA, USA). Values in the text are expressed as mean ± standard deviation (SD), and differences were considered significant at *p* < 0.05.

## 3. Results

### 3.1. Proliferation Effects of TUDCA on hMSCs

hMSCs were treated with TUDCA at the indicated concentrations, and cell proliferation was assessed daily. Overall, cell proliferation increased over time in a concentration-dependent manner. On 1 day and 3 days, the 500 μM TUDCA-treated group showed significantly increased proliferation compared with 0 μM TUDCA. After 5 days, cell proliferation was induced in a concentration-dependent manner with 10–500 μM TUDCA (Figure 1).

### 3.2. TUDCA-Induced Osteogenic Differentiation of hMSCs

The osteogenic differentiation effects of TUDCA on hMSCs were evaluated using ALP activity and calcium accumulation assays. Early differentiation was confirmed using an ALP activity assay, and ALP activity was statistically significantly higher at all concentrations, particularly at 500 μM TUDCA (Figure 2A). Mineralization was measured through calcium accumulation analysis to confirm final differentiation, which showed statistically significant high quantitative values of calcium accumulation at 250 and 500 μM TUDCA (Figure 2B). In addition, cell images were significantly stained in 500 μM TUDCA (Figure 2C).

### 3.3. Identification of Osteogenic Differentiation Markers in hMSCs Treated with TUDCA

As shown above, osteogenic differentiation was remarkably induced at 250 and 500 μM TUDCA in hMSCs. Therefore, we investigated osteogenic differentiation at the mRNA level at these two concentrations. The expression of osteogenic differentiation gene markers was determined using qRT-PCR. The expression of ALP, COL1A1, BSP, OCN, OPN, and OSX was significantly higher in 500 μM TUDCA treatment. The expression of ALP, COL1A1, BSP, OCN, and OPN was significantly high in 250 μM TUDCA treatment, but not OSX (Figure 3).

### 3.4. Identification of EGFR Signaling Pathway Genes in hMSCs Treated with TUDCA

The expression of genes involved in the signaling pathway of osteogenic differentiation induced by TUDCA in hMSCs was determined. As a result of searching the gene database (https://www.genecards.org, accessed on 27 May 2022) for specific genes identified as potential targets of TUDCA in hMSC, the EGFR gene was highly correlated. The results show that EGFR, p-Akt, and CREB1 were highly expressed at the mRNA and protein levels, and cyclin D1 and cyclin E1, which are target genes for this signaling pathway, were also significantly highly expressed (Figure 4). It has been suggested that TUDCA induces EGFR in hMSCs and osteogenic differentiation by regulating cyclin D1 and cyclin E1 through phosphorylating Akt in the cytoplasm and stimulating CREB1, a nuclear transcription factor.

### 3.5. EGFR Inhibitor-Induced Osteogenic Differentiation and Gene Expression in hMSCs Treated with TUDCA

To determine whether the EGFR plays a role in TUDCA-induced osteogenic differentiation of hMSCs, cells treated with a specific inhibitor (AG1478) of the EGFR were subjected to ALP activity and calcium accumulation measurements. ALP activity and calcium accumulation were significantly decreased in the group treated with 500 μM TUDCA together with an EGFR inhibitor compared with the group treated with 500 μM TUDCA alone. However, in the group treated with TUDCA 250 μM together with the EGFR inhibitor, ALP activity and calcium accumulation were slightly decreased compared with those in the group treated with 250 μM TUDCA, but the difference was not significant (Figure 5).

In addition, the gene expression of osteogenic differentiation markers, ALP, COL1A1, BSP, OPN, and OSX, at the mRNA level was significantly decreased in the group treated with 500 μM TUDCA together with the EGFR inhibitor compared with the group treated with 500 μM TUDCA alone, but not OCN. The expression of ALP, COL1A1, BSP, OCN, and OSX was significantly decreased in 250 μM TUDCA treatment, but not OPN (Figure 6A).

Our results show that the expression of signaling genes related to osteogenic differentiation, EGFR, p-Akt, and CREB1 also significantly decreased in the group treated with 500 μM TUDCA together with an EGFR inhibitor compared with the group treated with 500 μM TUDCA alone. In addition, the expression of cyclin D1 and cyclin E1, the target genes for signal transduction, was significantly reduced (Figure 6B). Therefore, we found that the EGFR plays an important role in the TUDCA-induced osteogenic differentiation of hMSCs.

## 4. Discussion

MSCs are pluripotent stromal cells and are the most used stem cells in current preclinical and clinical studies of skeletal diseases. In addition, TUDCA as a stem cell inducer has recently been used in research on bone regeneration. We investigated the bone regeneration effect by TUDCA in human mesenchymal stem cells and the signaling pathway through the expression of specific genes for the mechanism of osteogenic differentiation.

The proliferation of hMSCs increased as the concentration of TUDCA increased, indicating that there was no cytotoxicity. In addition, the result of the analysis of calcium accumulation and ALP activity shows that TUDCA induced bone differentiation. In particular, we showed that the osteogenic differentiation effect was remarkable after treatment with 250 and 500 μM TUDCA. Lower concentrations of TUDCA reportedly have this effect; we speculate that optimal concentrations are different in different cell lines.

Osteogenic differentiation was remarkably induced at 250 and 500 μM TUDCA, with significantly upregulated osteogenic differentiation gene markers at the mRNA level at these two concentrations. The results also show that the expression of COL1A1 was remarkably high. In contrast to other studies, increased expression of RUNX2 was not observed in this study. The expression decreased, suggesting that the osteogenic differentiation pathway induced by TUDCA is different from the physiological pathway. A previous study reported that drugs increased ALP activity and calcium accumulation but suppressed Runx2 expression [20]. Other studies have reported that it is not expressed due to differences in drug concentration [21]. Similarly, this study suggests that the mechanism of action of TUDCA in osteogenic differentiation is different from the existing mechanism of action. In addition, in this study, the EGFR was induced by TUDCA, and it was previously reported that the EGFR is closely related to COL1A1 [22]. Our results show that almost no expression of COL1A1 was observed after TUDCA and EGFR inhibitor treatment in this study (Figure 6A).

Genes related to TUDCA-induced osteogenic differentiation were searched in a gene database (https://www.genecards.org, accessed on 27 May 2022), and those with high relevance scores, namely EGFR and CREB1, were identified. These two genes are correlated with each other [23,24,25]. The EGFR is known to play an important role in osteogenic differentiation [26,27], and CREB1 is known to regulate cyclin D1 and cyclin E1 [27,28,29,30]. Based on these findings, this study was conducted based on the hypothesis that the EGFR is induced by TUDCA, and the nuclear CREB1 transcript reacts with Akt phosphorylation to regulate bone differentiation by regulating the target genes cyclin D1 and cyclin E1. We found that EGFR/p-Akt/CREB1 was induced by TUDCA in hMSCs (Figure 4). To validate this result, the EGFR/p-Akt/CREB1 pathway was evaluated after treatment with 10 μM of the EGFR inhibitor, AG1478. The expression of EGFR/p-Akt/CREB1 was reduced, as was the expression of cyclin D1 and cyclin E1. ALP activity and calcium accumulation by TUDCA were reduced by treatment with an EGFR inhibitor, and COL1A1 was particularly decreased among osteogenic differentiation marker genes (Figure 5 and Figure 6).

In this study, we demonstrated that TUDCA induced the differentiation of hMSCs into osteoblasts. It has been suggested that TUDCA-induced osteoblast differentiation induces differentiation by regulating cyclin D1 and cyclin E1 through the EGFR/p-Akt/CREB1 pathway. Based on this result, we suggest that it can contribute to the study of mesenchymal stem cell therapy required for bone regeneration in cases of bone defects. Additional in vivo studies are needed to determine whether bone formation occurs through the EGFR/p-Akt/CREB1 pathway in an in vivo model of bone defects.

## 5. Conclusions

TUDCA induced the osteogenic differentiation of hMSCs. In hMSCs, stimulation by TUDCA induced EGFR, and CREB1 responded by phosphorylating Akt. Osteogenic differentiation occurred via this step by regulating cyclin D1 and cyclin E1. Therefore, TUDCA may induce osteogenic differentiation via the EGFR/p-Akt/CREB1 pathway.

## Figures and Tables

**Figure 1 cells-12-01463-f001:**
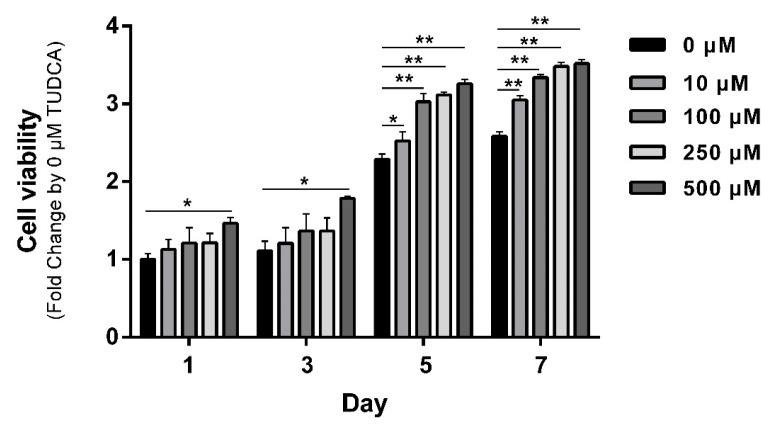
Proliferation of hMSCs induced by TUDCA. hMSCs were treated with TUDCA at concentrations of 10, 100, 250, and 500 μM, and cell proliferation was evaluated daily using water-soluble tetrazolium salt (WST) -1 assays. A microplate reader was used to measure the absorbance at 450 nm. All data represent the mean ± S.D. of five independent experiments. * *p* < 0.05, ** *p* < 0.001 versus the control group.

**Figure 2 cells-12-01463-f002:**
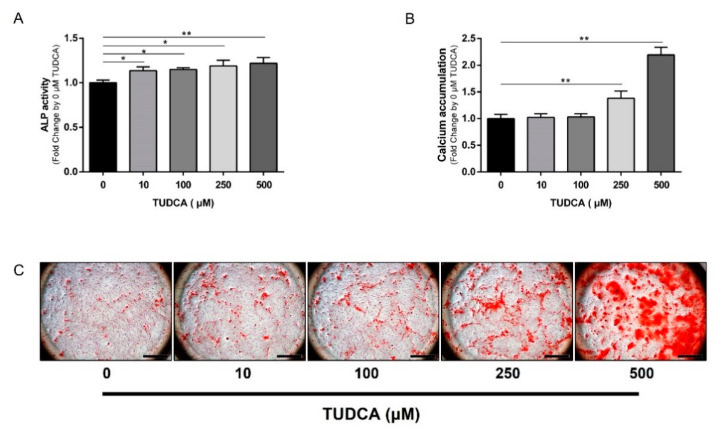
Osteogenic differentiation of hMSCs induced by TUDCA. hMSCs were treated with TUDCA at concentrations of 10, 100, 250, and 500 μM. (**A**) ALP activity assay results on day 7 after TUDCA treatment. (**B**) Calcium accumulation values of cells stained with AR-S solution 20 d after TUDCA treatment. (**C**) Images of cells stained with AR-S solution. Scale bars = 1 mm. All data represent the mean ± S.D. of five independent experiments. * *p* < 0.05, ** *p* < 0.001 versus the control group.

**Figure 3 cells-12-01463-f003:**
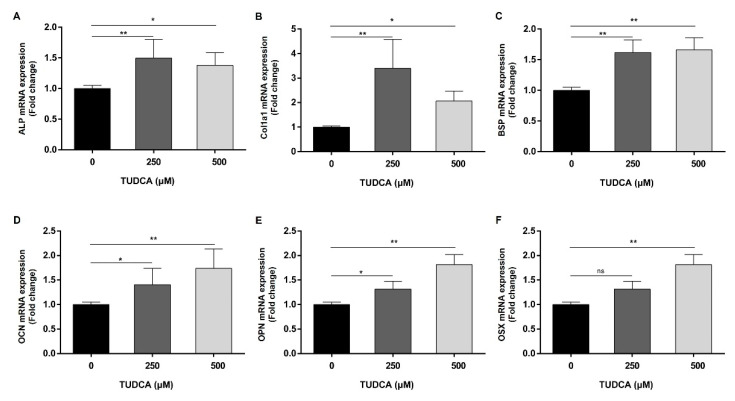
Identification of osteogenic differentiation markers in hMSCs treated with TUDCA. hMSCs were treated with TUDCA at concentrations of 250 and 500 μM, total RNA was extracted from TUDCA-treated hMSCs, and qRT-PCR was performed. (**A**) ALP, (**B**) Col1a1, (**C**) BSP, (**D**) OCN, (**E**) OPN, (**F**) OSX. Values were normalized to GAPDH mRNA. mRNA levels of hMSC osteogenic differentiation markers were determined. All data represent the mean ± S.D. of five independent experiments. * *p* < 0.05, ** *p* < 0.001 versus the control group.

**Figure 4 cells-12-01463-f004:**
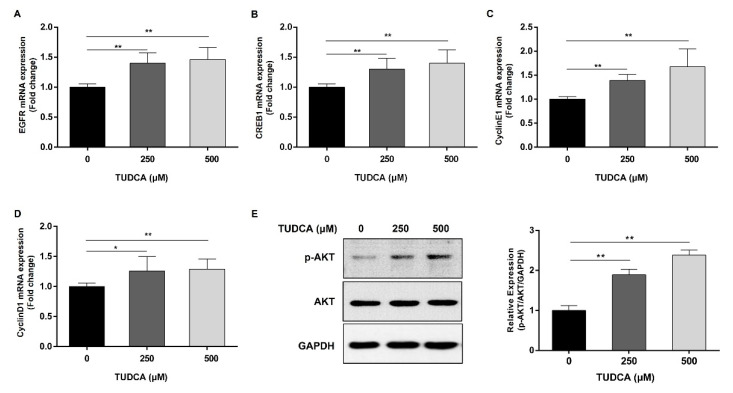
Identification of EGFR signaling pathway genes in hMSCs treated with TUDCA. hMSCs were treated with TUDCA at concentrations of 250 and 500 μM, total RNA and protein were extracted from the cells, and then qRT-PCR and Western blotting were performed. (**A**–**D**) The mRNA levels of EGFR, CREB1, cyclin D1, and cyclin E1 were determined and normalized to GAPDH mRNA. (**E**) Protein levels of p-Akt, Akt, and GAPDH were determined. All data represent the mean ± S.D. of five independent experiments. * *p* < 0.05, ** *p* < 0.001 versus the control group.

**Figure 5 cells-12-01463-f005:**
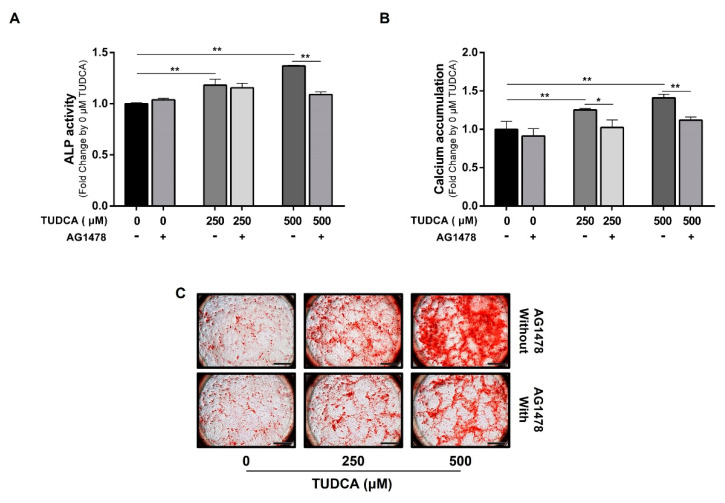
Inhibition of TUDCA-induced osteogenic differentiation of hMSCs by an EGFR inhibitor (AG1478). hMSCs were pretreated with AG1478, an EGFR inhibitor, and then treated with 250 and 500 μM TUDCA, and each experiment was performed by time. (**A**) ALP activity after 7 d; (**B**,**C**) calcium accumulation after 20 d were determined, and an image of cells stained with AR-S solution is shown. Scale bars = 1 mm. All data represent the mean ± S.D. of five independent experiments. * *p* < 0.05, ** *p* < 0.001 versus the control group.

**Figure 6 cells-12-01463-f006:**
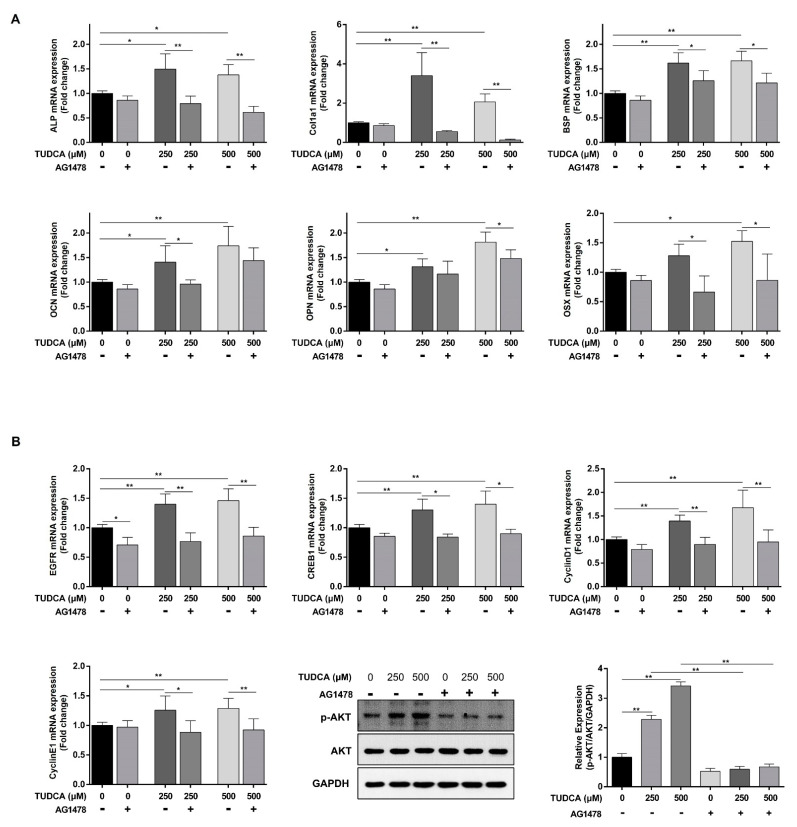
EGFR inhibitor-induced osteogenic differentiation and gene expression in hMSCs treated with TUDCA. (**A**) Total RNA was extracted from cells and expression of osteogenic differentiation marker genes was determined by qRT-PCR; values were normalized to GAPDH mRNA. (**B**) mRNA levels of EGFR, CREB1, cyclin D1, and cyclin E1 were determined and normalized to GAPDH mRNA. The protein levels of p-Akt, Akt, and GAPDH were determined. All data represent the mean ± S.D. of five independent experiments. * *p* < 0.05, ** *p* < 0.001 versus the control group.

## Data Availability

All data and materials generated or used during the study are available from the corresponding authors upon reasonable request.

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
