# Peer review of "Tauroursodeoxycholic Acid Enhances Osteogenic Differentiation through EGFR/p-Akt/CREB1 Pathway in Mesenchymal Stem Cells"

_cells, 2023, doi:10.3390/cells12111463_

Round 1
Reviewer 1 Report
In this study, Kang et al. investigated the effect of tauroursodeoxycholic acid (TUDCA) on osteogenesis of human mesenchymal stem cells (hMSCs). The authors showed a dramatic effect of TUDCA on osteogenesis of hMSCs. Additionally, the authors identified the involvement of EGFR-signaling in TUDCA-mediated osteogenesis of hMSCs. This is an interesting study with the potential to contribute to the research field of bone regeneration. However, there are a couple of concerns that need to be addressed.
1. It was surprising that the authors only briefly mentioned EGFR signaling as a target of TUDCA in their Discussion (lines 220-224) without clearly describing the identification process in the Results section. Therefore, I recommend that the authors provide a clear and detailed description of the following points in the Results section: 1) The experiments that were performed to identify candidate targets for TUDCA in hMSCs, 2) The specific genes that were identified as potential targets, 3) The criteria that were used to ultimately select EGFR as a target for further investigation. By providing a thorough explanation of these points, readers will have a better understanding of the authors' rationale for investigating EGFR signaling and its role in TUDCA-mediated osteogenesis of hMSCs.
2. If the EGFR inhibitor AG1478 inhibited the effect of TUDCA, it would suggest that TUDCA induces the activation of EGFR. However, this study did not examine or confirm this point. It is important to perform experiments that TUDCA-treatment activated EGFR in hMSCs by detecting EGFR phosphorylation, and EGFR inhibitor AG1478 inhibited its phosphorylation. Conducting these experiments would provide convincing evidence for the involvement of EGFR signaling in TUDCA-mediated osteogenesis of hMSCs.
3. The authors should include the results of Runx2 expression in their study (as mentioned in lines 212-213), and address the discrepancy between their findings and those of previous studies. Additionally, they should provide appropriate references to support the previous studies they are referring to.
This manuscript needs extensive English language editing.
Author Response
Thank you for reviewing our manuscript and for providing valuable suggestions.
Point 1: It was surprising that the authors only briefly mentioned EGFR signaling as a target of TUDCA in their Discussion (lines 220-224) without clearly describing the identification process in the Results section. Therefore, I recommend that the authors provide a clear and detailed description of the following points in the Results section: 1) The experiments that were performed to identify candidate targets for TUDCA in hMSCs, 2) The specific genes that were identified as potential targets, 3) The criteria that were used to ultimately select EGFR as a target for further investigation. By providing a thorough explanation of these points, readers will have a better understanding of the authors' rationale for investigating EGFR signaling and its role in TUDCA-mediated osteogenesis of hMSCs.
Response 1: Thanks for your important comments. It has been revised and supplemented according to your comment. The revised content is as follows and is shown in the original manuscript. (lines 168-171)
The expression of genes involved in the signaling pathway of osteogenic differentiation induced by TUDCA in hMSCs was determined. As a result of searching the gene database (https://www.genecards.org) for specific genes identified as potential targets of TUDCA in hMSC, the EGFR gene was found to be highly correlated.
Point 2: If the EGFR inhibitor AG1478 inhibited the effect of TUDCA, it would suggest that TUDCA induces the activation of EGFR. However, this study did not examine or confirm this point. It is important to perform experiments that TUDCA-treatment activated EGFR in hMSCs by detecting EGFR phosphorylation, and EGFR inhibitor AG1478 inhibited its phosphorylation. Conducting these experiments would provide convincing evidence for the involvement of EGFR signaling in TUDCA-mediated osteogenesis of hMSCs.
Response 2: We agree that It is important to perform experiments that TUDCA-treatment activated EGFR in hMSCs by detecting EGFR phosphorylation, and EGFR inhibitor AG1478 inhibited its phosphorylation. However, other studies have explained its mechanism of action by gene expression of EGFR without showing phosphorylation of EGFR. It has been shown that increased gene expression of EGFR activates its mechanism of action, and it has also been reported that the EGFR inhibitor AG1478 reduces the phosphorylation of other proteins in the down-path, thereby reducing the expression of target genes [1-3]. EGFR phosphorylation has been reported to be undetected.
Similarly, in this study, it was confirmed that EGFR gene expression was increased and the mechanism of action was activated by TUDCA, and the EGFR inhibitor AG1478 inhibited the action of TUDCA, reducing EGFR gene expression and downstream pathway, namely p-Akt. Therefore, it was suggested that TUDCA induces EGFR activation in hMSC.
Point 3: The authors should include the results of Runx2 expression in their study (as mentioned in lines 212-213), and address the discrepancy between their findings and those of previous studies. Additionally, they should provide appropriate references to support the previous studies they are referring to.
Response 3: I have added content and references based on your comments.(line 229-231)
A previous study reported that drugs increased ALP activity and calcium accumulation but suppressed Runx2 expression [4].
References
- Ma L, Yan H, Zhou Q. AG1478 inhibits the migration and invasion of cisplatin‑resistant human lung adenocarcinoma cells via the cell cycle regulation by matrix metalloproteinase‑9. Oncol Lett. 2014;8(2):921-7.
- De Almeida V, de Melo A, Meira D, Pires A, Nogueira-Rodrigues A, Pimenta-Inada H, Alves F, Moralez G, Thiago L, Ferreira C. Radiotherapy modulates expression of EGFR, ERCC1 and p53 in cervical cancer. Braz J Med Biol Res. 2017;51.
- Mialon A, Sankinen M, Söderström H, Junttila TT, Holmström T, Koivusalo R, Papageorgiou AC, Johnson RS, Hietanen S, Elenius K. DNA topoisomerase I is a cofactor for c-Jun in the regulation of epidermal growth factor receptor expression and cancer cell proliferation. Mol Cell Biol. 2005;25(12):5040-51.
- Ding J, Ghali O, Lencel P, Broux O, Chauveau C, Devedjian J, Hardouin P, Magne D. TNF-α and IL-1β inhibit RUNX2 and collagen expression but increase alkaline phosphatase activity and mineralization in human mesenchymal stem cells. Life Sci. 2009;84(15-16):499-504.
Reviewer 2 Report
In the presented manuscript the Authors describe an influence of tauroursodeoxycholic acid (TUDCA) on the process of osteogenic differentiation of human bone marrow-derived MSC.
The experiments were well performed but I have some remarks:
1. The proliferation assay was not described in Material and Method section, the only short description is in Figure 1 legend; however, the Y axis in this figure is titled as “fold” but it is not written what does it mean (fold of what?).
2. In the Results section there is a description of results depicted in Figure 1 but it is stated only that “the increase was statistically significant at TUDCA 500 uM on 1day”. What about other days and concentrations? According to the figure the statistical significance is also present on other days and concentrations of TUDCA.
3. There is some mess in the description of the results depicted in Figure 2, lines 129-135. I do not understand the description. What is Deep Red? Again, there is a question of “fold” on Y axis, the same in Figure 5.
4. The last remark deals with References. There is a mess in journal citation: sometimes it is an abbreviation of the journal name, sometimes a full name is given.
Author Response
Thank you for reviewing our manuscript and for providing valuable suggestions.
Point 1: The proliferation assay was not described in Material and Method section, the only short description is in Figure 1 legend; however, the Y axis in this figure is titled as “fold” but it is not written what does it mean (fold of what?).
Response 1: Based on your comments, we have added the contents of the cell proliferation assay to the Materials and Methods section, and the contents are shown below. (lines 69-77)
Cell proliferation assay
Cells were seeded at a density of 5 × 103 cells/well in a 96-well plate and treated with TUDCA at different concentrations. After culturing for days 1, 3, 5, and 7, the cells were washed twice with phosphate buffered saline. EZ-cytox™ (Daeil Lab Service, Seoul, Korea), which has a method of analyzing cell proliferation by measuring the water-soluble te-trazolium salts (WST-1) reacted by dehydrogenase in mitochondria, is added to 10% of the culture medium and treated with cells and reacted for 1 hour in a 37°C incubator. After the reaction, the culture medium was analyzed by a microplate reader (SpectraMax M3; Mo-lecular Devices, Sunnyvale, CA, USA) was used to measure the absorbance at 450 nm.
In the figure, the title of the Y-axis has been modified.
'Fold change by 0 μM TUDCA'
Point 2: In the Results section there is a description of results depicted in Figure 1 but it is stated only that “the increase was statistically significant at TUDCA 500 uM on 1day”. What about other days and concentrations? According to the figure the statistical significance is also present on other days and concentrations of TUDCA.
Response 2: Thank you very much for your comments. In accordance with your comments, the content was corrected and added to the manuscript, and the content is shown below. (Lines 131-134)
Overall, Cell proliferation increased over time in a concentration-dependent manner. On 1day and 3day, the 500 μM TUDCA-treated group showed significantly increased proliferation compared to 0 μM TUDCA. After 5 day, cell proliferation was induced in a concentration-dependent manner with 10-500 μM TUDCA (Fig. 1).
Point 3: There is some mess in the description of the results depicted in Figure 2, lines 129-135. I do not understand the description. What is Deep Red? Again, there is a question of “fold” on Y axis, the same in Figure 5.
Response 3: Thank you very much for your comments. In accordance with your comments, the content was corrected and added to the manuscript, and the content is shown below. (Lines 143-148)
The osteogenic differentiation effects of TUDCA on hMSCs was evaluated using ALP activity and calcium accumulation assays. Early differentiation was confirmed using an ALP activity assay, and ALP activity was statistically significantly higher at all concentrations, particularly at 500 μM TUDCA (Fig. 2A). Mineralization was measured through calcium accumulation analysis to confirm final differentiation, which showed statistically significant high quantitative values of calcium accumulation at 250 and 500 μM TUDCA (Fig. 2B). In addition, cell images were strongly stained in 500 μM TUDCA (Fig. 2C).
In the figure, the title of the Y-axis has been modified.
'Fold change by 0uM TUDCA'
Point 4: The last remark deals with References. There is a mess in journal citation: sometimes it is an abbreviation of the journal name, sometimes a full name is given.
Response 4: We sincerely apologize for the confusion caused by the journal citation. The journal name was abbreviated and indicated.
Example
before modification - Gao F, Chiu S, Motan D, Zhang Z, Chen L, Ji H, Tse H, Fu Q-L, Lian Q. Mesenchymal stem cells and immunomodulation: current status and future prospects. Cell death & disease. 2016;7(1):e2062-e.
after modification - Gao F, Chiu S, Motan D, Zhang Z, Chen L, Ji H, Tse H, Fu Q-L, Lian Q. Mesenchymal stem cells and immunomodulation: current status and future prospects. Cell Death Dis. 2016;7(1):e2062-e.
Round 2
Reviewer 1 Report
The authors answered to my questions. They explained carefully, and I'm satisfied.